# miR-542-3p Contributes to the HK2-Mediated High Glycolytic Phenotype in Human Glioma Cells

**DOI:** 10.3390/genes12050633

**Published:** 2021-04-23

**Authors:** Junhyung Kim, Min Woo Park, Young Joon Park, Ju Won Ahn, Jeong Min Sim, Suwan Kim, Jinhyung Heo, Ji Hun Jeong, Mihye Lee, Jaejoon Lim, Jong-Seok Moon

**Affiliations:** 1Department of Integrated Biomedical Science, Soonchunhyang Institute of Medi-bio Science (SIMS), Soonchunhyang University, Chungcheongnam-do, Cheonan 31151, Korea; hotdog58@naver.com (J.K.); pmw0269@sch.ac.kr (M.W.P.); jihun@sch.ac.kr (J.H.J.); mihyelee@sch.ac.kr (M.L.); 2Bundang CHA Medical Center, Department of Neurosurgery, CHA University, Yatap-dong 59, Seongnam 13496, Korea; yjparkcb@gmail.com (Y.J.P.); dicendice@gmail.com (J.W.A.); simti123@naver.com (J.M.S.); suwankimm@chauniv.ac.kr (S.K.); 3Bundang CHA Medical Center, Department of Pathology, CHA University, Yatap-dong 59, Seongnam 13496, Korea; sacrum77@chamc.co.kr

**Keywords:** miR-542-3p, cellular proliferation, hexokinase 2, glycolysis, glioma

## Abstract

(1) Background: The elevation of glucose metabolism is linked to high-grade gliomas such as glioblastoma multiforme (GBM). The high glycolytic phenotype is associated with cellular proliferation and resistance to treatment with chemotherapeutic agents in GBM. MicroRNA-542-3p (miR-542-3p) has been implicated in several tumors including gliomas. However, the role of miR-542-3p in glucose metabolism in human gliomas remains unclear; (2) Methods: We measured the levels of cellular proliferation in human glioma cells. We measured the glycolytic activity in miR-542-3p knockdown and over-expressed human glioma cells. We measured the levels of miR-542-3p and HK2 in glioma tissues from patients with low- and high-grade gliomas using imaging analysis; (3) Results: We show that knockdown of miR-542-3p significantly suppressed cellular proliferation in human glioma cells. Knockdown of miR-542-3p suppressed HK2-induced glycolytic activity in human glioma cells. Consistently, over-expression of miR-542-3p increased HK2-induced glycolytic activity in human glioma cells. The levels of miR-542-3p and HK2 were significantly elevated in glioma tissues of patients with high-grade gliomas relative to that in low-grade gliomas. The elevation of HK2 levels in patients with high-grade gliomas were positively correlated with the high levels of miR-542-3p in GBM and low-grade gliomas (LGG) based on the datasets from the Cancer Genome Atlas (TCGA) database. Moreover, the high levels of miR-542-3p were associated with poor survival rate in the TCGA database; (4) Conclusions: miR-542-3p contributes to the HK2-mediated high glycolytic phenotype in human glioma cells.

## 1. Introduction

Glioma is a type of malignant brain tumor [1,2]. Based on the WHO brain tumor classification criteria, gliomas are divided into those with biologically benign features (grade I and II as low-grade glioma (LGG)) and those with diffusely infiltrating features (grade III and IV as high-grade glioma) by histological pathologic evaluation and genetic molecular patterns [3]. High-grade glioma including glioblastoma multiforme (GBM) is the most aggressive type of glioma [4]. High-grade glioma (grade III and IV) exhibits poor prognosis with a high recurrence rate [4,5].

In various types of cancer, glucose metabolism is important for the maintenance of cellular proliferation, growth, and homeostasis [6,7]. GBM also shows the activation of glucose metabolism as a main source of energy production including ATP [8]. As an important metabolic pathway, glycolysis is critical for energy metabolism, cellular proliferation, and survival [9]. The elevation of glycolysis has been linked to high-grade glioma such as GBM [9,10]. In the glycolysis pathway, hexokinase (HK) is the first key enzyme [11]. HK phosphorylates glucose to produce glucose-6-phosphate (G6P) [8]. HK has a role in the initiation of glucose utilization and glucose-mediated cellular metabolisms [12]. There are four highly homologous hexokinase isoforms (HK1, HK2, HK3, and HK4 (also known as glucokinase)) in mammalian cells [13]. As a key enzyme in the glycolysis pathway, HK2 enhances the growth of tumors in GBM [8]. In the final step of the glycolysis pathway, lactate dehydrogenase (LDH) produces lactate through the conversion of pyruvate to lactate [14,15,16]. In mammalian cells, LDH is composed of four subunits. LDHA (the M subunit of LDH as a muscle type) and LDHB (the H subunit of LDH as a heart type) have the same active sites and amino acids, which participate in the enzyme reaction [16,17]. However, the mechanism for the regulation of the HK2-dependent high glycolytic phenotype in high-grade glioma is still unclear.

MicroRNAs (miRNAs) are small single-stranded non-coding RNAs [18,19]. miRNAs have a role in RNA silencing and post-transcriptional regulation of target gene expression by base-pairing with partially or fully complementary sequences [18,19]. miRNAs can affect various biological processes including cell proliferation, cell survival, differentiation, and cellular metabolism [20,21]. In cancer, various miRNAs play a role as tumor suppressors or oncogenes [20,21]. Among various miRNAs, the role of miR-542-3p has been implicated in various tumors including astrocytoma, neuroblastoma, breast cancer, and colorectal cancer [22,23,24,25,26]. However, the role of miR-542-3p in the high glycolytic phenotype in high-grade glioma remains unclear.

Here, we show that the knockdown of miR-542-3p suppressed cellular proliferation in human glioma cells. Knockdown of miR-542-3p suppressed HK2-mediated glycolytic activity via the reduction of HK2 expression. Consistently, the over-expression of miR-542-3p increased HK2-mediated glycolytic activity. The levels of miR-542-3p and HK2 were significantly elevated in glioma tissues of patients with high-grade gliomas compared to those with low-grade gliomas. The levels of the *HK2* gene were positively correlated with the high levels of miR-542-3p in the GBM and LGG datasets from the Cancer Genome Atlas (TCGA) database. Moreover, the high levels of miR-542-3p were associated with poor prognosis in patients with gliomas using analysis of the TCGA database. Our results suggest that miR-542-3p contributes to the HK2-mediated high glycolytic phenotype in human glioma cells.

## 2. Materials and Methods

### 2.1. Human Study

This human subject study was performed in accordance with the Helsinki Declaration. The human study protocol was reviewed and approved by the Institutional Review Board (IRB) of Soonchunhyang University Hospital Cheonan (IRB number: SCHCA 2020-03-030-001) and Bundang CHA medical center (IRB number: 2016-04-012). All the patients provided written informed consent before entry into the study. The current human study included patients with low-grade gliomas (grade I, *n* = 2/grade II, *n* = 3) and high-grade gliomas (grade III, *n* = 3/grade IV, *n* = 3) (Appendix A).

### 2.2. Reagents

The immunoblot analysis was conducted with the following primary antibodies: monoclonal rabbit anti-HK2 antibody (#2024, Cell Signaling Technology, Danvers, MA, USA), monoclonal rabbit anti-LDH-A antibody (#3582, Cell Signaling Technology, Danvers, MA, USA), monoclonal rabbit anti-PFKP antibody (#8164, Cell Signaling Technology, Danvers, MA, USA), monoclonal rabbit anti-PKM2 antibody (#4053, Cell Signaling Technology, Danvers, MA, USA), and monoclonal mouse anti-β-actin (A5316, Sigma-Aldrich, St. Louis, MO, USA). For nuclear staining and mounting in immunofluorescence analysis, tissue slides were processed with Fluoroshield™ with DAPI (F6057, Sigma-Aldrich, St. Louis, MO, USA). For mounting in the immunohistochemistry analysis, tissues were mounted onto gelatin-coated slides and were processed with Canada Balsam (Wako, Tokyo, Japan) following the dehydration of sections.

### 2.3. Human Glioma Cells

Human glioma U87MG cells (ATCC^®^ HTB-14™, ATCC, Manassas, VA, USA) were used. For the experiments with U87MG cells, U87MG cells were cultured in Dulbecco’s Modified Eagle Medium (DMEM) (11995065, Thermo Fisher Scientific, Waltham, MA, USA) containing 10% (*v*/*v*) fetal bovine serum (FBS), 100 units/mL penicillin, and 100 mg/mL streptomycin (A1261301, Thermo Fisher Scientific, Waltham, MA, USA). In the initial step of transfection experiments, the efficiency of transfection was validated in U87MG cells treated with GFP-expressing plasmid (pCMV6-AC-GFAP mammalian expression vector, Cat. No. PS100010, Origene, Rockville, MD, USA) as a positive control for transfection using lipofectamine LTX with Plus reagent (15338100, Thermo Fisher Scientific, Waltham, MA, USA) according to the manufacturer’s instructions. In addition, the effects of transfection reagents were identified in U87MG cells treated with only lipofectamine LTX with Plus reagent (15338100, Thermo Fisher Scientific, Waltham, MA, USA) as a mock control. The transfection reagents had no apparent effect in U87MG cells. In the experiments for the overexpression of human miR-542-3p, U87MG cells were seeded (2 × 10^5^ cells/6-well cell culture plates) and transfected with microRNA hsa-miR-542-3p (ugugacagauugauaacugaaa) (Dharmacon, Lafayette, CO, USA) or miRIDIAN microRNA mimic negative control #1 (Cat.no. CN-001000-01-20, Dharmacon) using lipofectamine LTX with Plus reagent (15338100, Thermo Fisher Scientific, Waltham, MA, USA) according to the manufacturer’s instructions. The effects of human miR-542-3p overexpression were analyzed at 24 h after transfection of 10 nM mimic or negative control in U87MG cells. For knockdown of human miR-542-3p, U87MG cells were seeded (2 × 10^5^ cells/6-well cell culture plates) and transfected with miRIDIAN microRNA human hsa-miR-542-3p-hairpin inhibitor (Cat.no. IH-300866-05-0005, Dharmacon, Lafayette, CO, USA) or miRIDIAN microRNA hairpin inhibitor negative control (Cat.no. IN-001005-01-20, Dharmacon) using Lipofectamine LTX with Plus reagent (15338100, Thermo Fisher Scientific, Waltham, MA, USA) according to the manufacturer’s instructions. The effects of human miR-542-3p knockdown were analyzed at 24 h after transfection of 10 nM mimic or negative control in U87MG cells. miRIDIAN microRNA hairpin inhibitor negative control had no apparent effect on the levels of HK2 in U87MG cells. Cellular morphology and the lengths of the cell bodies were analyzed by EVOS M5000 Imaging System according to the manufacturer’s instructions. (Thermo Fisher Scientific, Waltham, MA, USA).

### 2.4. 2 D and 3D Immunofluorescence and Morphology Analysis

In the immunofluorescence analysis with tissues, brain tissues with a thickness of 4 μm were sectioned and processed from paraffin embedded tissue blocks. After the deparaffinization, tissue slides were permeabilized in 0.5% Triton-X (T8787, Sigma-Aldrich, St Louis, MO, USA) at 25 °C for 10 min. Tissue slides were blocked with CAS-Block™ Histochemical Reagent (008120, Thermo Fisher Scientifi, Waltham, MA, USA). Next, tissue slides were incubated with primary antibody using monoclonal rabbit anti-HK2 antibody (#2024, Cell Signaling Technology, Danvers, MA, USA) (1:100) at 4 °C for 16 h. After the washing steps, tissue slides were incubated with secondary antibody using goat anti-rabbit IgG (H + L) Alexa Fluor 488 (A11008, Thermo Fisher Scientific, Waltham, MA, USA) (1:100) at 25 °C for 2 h. For nuclear staining, tissue slides were processed with Fluoroshield™ with DAPI (F6057, Sigma-Aldrich, St Louis, MO, USA). Two-dimensional and 3D immunofluorescence image analysis was conducted with THUNDER Imager Tissue (Leica Microsystems Ltd., Wetzlar, Germany). Two-dimensional and 3D immunofluorescence images from stained tissue slides were analyzed and quantified by LAS X image-processing software (Leica Microsystems Ltd., Wetzlar, Germany) and ImageJ software v1.52a (Bethesda, MD, USA). To ensure objectivity, all analyses were conducted with blinded conditions by two observers who performed analyses using identical conditions per experiment. For cellular morphologic analysis, cells were analyzed by EVOS M5000 Imaging System (Thermo Fisher Scientific, Waltham, MA, USA). The images were quantified by ImageJ software v1.52a (Bethesda, MD, USA).

### 2.5. In Situ Hybridization for Human miR-542-3p

Custom designed probes for human hsa-miR-542-3p were obtained from hsa-miR-542-3p miRCURY LNA miRNA detection probe (Product No. 339111, Cat. No. YD00612610-BCD, QIAGEN, Hilden, Germany) following the manufacturer’s instructions (miRCURY LNA miRNA Detection probes Handbook). Briefly, hybridization was conducted by a protocol of miRCURY LNA miRNA ISH buffer and controls (Cat No./ID: 339459, QIAGEN, Hilden, Germany). Hybridization was performed with an miRNA in situ hybridization probe (12.5 nM) at 25 °C for 1 h. Tissue sections were washed with 1 × PBS followed by the hybridization step. Signal amplification was performed by the branched DNA technology with a series of sequential hybridization and washing steps. After hybridization, tissue slides were incubated with blocking solution at 25 °C for 15 min. After incubation with Anti-DIG reagent at 25 °C for 15 min, sections were treated with an alkaline-phosphatase (AP)-labeled probe at 30 °C for 2 h and then an AP enhancer at 25 °C for 5 min. Slides were incubated with KTBT buffer at 25 °C for 5 min. Fast red nuclear substrate stainings were conducted at 40 °C for 30 min. After the stained tissue slides were fixed, they were mounted using Eukitt^®^ Quick-hardening mounting medium (03989, Sigma-Aldrich, St Louis, MO, USA). Stained tissue slides were observed and analyzed with an Olympus BX53M microscope. The images were analyzed and quantified by Olympus Stream software and ImageJ software v1.52a (Bethesda, MD, USA).

### 2.6. Immunoblot Analysis

After the collection of cells, cells were lysed with NP40 Cell Lysis Buffer (FNN0021, Thermo Fisher Scientific, Waltham, MA, USA). Lysates from cells were centrifuged at 15,300× *g* at 4 °C for 10 min. The proteins in the supernatants were isolated. For the measurement of protein concentrations, a Bradford assay kit (500-0006, Bio-Rad Laboratories, Hercules, CA, USA) was used. The proteins were electrophoresed on NuPAGE 4–12% Bis-Tris gels (Thermo Fisher Scientific, Waltham, MA, USA). For the incubation step of the primary antibody, the separated proteins in gels were transferred to Protran nitrocellulose membranes (10600001, GE Healthcare Life science, Pittsburgh, PA, USA). After washing the membranes with TBS-T (TBS (170-6435, Bio-Rad Laboratories) and 1% (*v*/*v*) Tween-20 (170-6531, Bio-Rad Laboratories) at 25 °C for 5 min, the membranes were incubated with the blocking buffer using 5% (*w*/*v*) bovine serum albumin (BSA) (9048-46-8, Santa Cruz Biotechnology, Dallas, TX, USA) in TBS-T at 25 °C for 30 min. After the washing steps for the membranes with TBS-T, the membranes were incubated with primary antibody (1:1000) diluted in 1% (*w*/*v*) BSA in TBS-T at 4 °C for 16 h. After the incubation with the primary antibody, membranes were incubated with the secondary antibody using the horseradish peroxidase (HRP)-conjugated goat anti–rabbit IgG–HRP (sc-2004) (1:2500) (Santa Cruz Biotechnology) and the horseradish peroxidase (HRP)-conjugated goat anti–mouse IgG–HRP (sc-2005) (1:2500) (Santa Cruz Biotechnology, Dallas, TX, USA) diluted in TBS-T at 25 °C for 1 h. After all steps, the specific immunoreactive bands were detected by SuperSignal West pico chemiluminescent substrate (34078, Thermo Scientific, Waltham, MA, USA).

### 2.7. Glycolysis Activity Assay

For the glycolytic activity assay, human glioma U87MG cells were seeded and cultured in XF96e cell culture microplates (101085-004, Agilent Technologies, Inc., Santa Clara, CA, USA). U87MG cells (5 × 10^4^ cells/well) were used in the experiments of miR-542-3p knockdown. In the experiments of miR-542-3p overexpression, U87MG cells (2.5 × 10^4^ cells/well) were used. For the measurement of glycolytic flux and activity, the ECAR levels were analyzed by a Seahorse XF96e bioanalyzer using the XF glycolysis stress test kit (102194-100, Agilent Technologies, Inc., Santa Clara, CA, USA) following the manufacturer’s instructions. The ECAR levels were monitored in a time-dependent manner. The ECAR levels were measured in U87MG cells treated with glucose (10 mM), oligomycin (2 μM), and 2-deoxyglucose (2DG) (10 mM), respectively.

### 2.8. Cell Proliferation Assay

The cellular proliferation assay was conducted with human glioma cells (2 × 10^4^ cells/well in 96-well cell culture plates) in time-dependent manner. The levels of cellular proliferation were measured by an MTS assay kit (ab197010, Abcam, Cambridge, UK) using a colorimetric method using absorbance at 490 nm for the sensitive quantification of viable cells according to the manufacturer’s instructions.

### 2.9. Analysis of the TCGA Dataset

We downloaded RNA-seq and survival data of lower grade glioma (LGG) and glioblastoma (GBM) from the Xena TCGA database (https://xenabrowser.net/, accessed date: From 11 December 2020 to 15 December 2020). In total, 689 patients participated in this study. Among the data, there were grade II, III patients (*n* = 25,5269) and grade IV patients (*n* = 169). Expression values of genes were transformed to a 0–1 standard scale and merged with survival and pathological grade data. The survival data showed overall survival time and events. This process was performed by Python 2.7.

### 2.10. Statistical Analysis

In the statistical analyses of the experiments, both the assumptions of normality and homogeneity of variance were evaluated. For normal distribution, a Shapiro–Wilk test was performed. For the homogeneity of variance, Levene’s test was conducted. Data were random, independent, and normally distributed. Data had a common variance. All data are presented as mean ± standard deviation (SD) or standard error of the mean (SEM). For the comparison of two groups, statistical analysis was evaluated by a two-tailed Student’s *t*-test. For the comparison of multiple groups, statistical analysis was evaluated by analysis of variance (ANOVA) (with post hoc comparisons using Dunnett’s test) using a statistical software package (GraphPad Prism version 8.0, GraphPad Software Inc., San Diego, CA, USA). To determine statistically different of gene expression between each neoplasm histologic grade, one-sided and two-sided *t*-tests or Wilcoxon tests were performed on the TCGA and NanoString datasets, respectively. Pearson’s correlation analysis was used for evaluating the association of expression between two genes. Linear regression was performed to confirm the relationship between miR-542-3p and HK2 expression level. Kaplan–Meier estimation and Cox-regression analysis were performed to analyze the prognosis of genes with overall survival data. Concordance index was calculated to provide accuracy with survival proportion. Kaplan–Meier estimation was performed to analyze prognosis of genes with overall survival data. All statistical analyses were calculated by R studio (Version: 1.1.456). *p* values *, *p* < 0.05, **, *p* < 0.01, and *** *p* < 0.001 were considered statistically significant.

## 3. Results

### 3.1. Knockdown of miR-542-3p Suppresses Cellular Proliferation in Human Glioma Cells

We investigated the role of miR-542-3p in human glioma. We examined whether the knockdown of miR-542-3p could suppress cellular proliferation in human glioma cells. We first analyzed the morphologic changes by miR-542-3p knockdown in human glioma cells. Knockdown of miR-542-3p (miR-542-3p KO) resulted in the reduction of cell body size and the condensation of cells compared to that in cells treated with microRNA hairpin inhibitor negative control (Control) (Figure 1A). The length of the cell body was significantly decreased by miR-542-3p knockdown relative to that in control (Figure 1B). Since miR-542-3p knockdown affected the cellular growth of human glioma cells, we investigated whether miR-542-3p knockdown could regulate the cellular proliferation in human glioma cells. Notably, knockdown of miR-542-3p significantly suppressed the cellular proliferation in a time-dependent manner compared to that in control (Figure 1C). These results suggest the knockdown of miR-542-3p suppresses cellular proliferation in human glioma cells.

### 3.2. miR-542-3p Regulates the HK2-Mediated High Glycolytic Phenotype in Human Glioma Cells

Next, we investigated the mechanisms by which miR-542-3p regulates cellular proliferation in glioma. Since the high-glycolytic phenotype is linked to cellular proliferation, we examined whether the knockdown of miR-542-3p could regulate the high glycolytic phenotype in human glioma cells. First, we analyzed the effects of miR-542-3p knockdown on the glycolytic activity in human glioma cells by the measurement of the extracellular acidification rate (ECAR) as a parameter of glycolytic activity (Figure 2A). The glycolytic activity was measured by the sequential treatment of glucose, oligomycin (a selective inhibitor for mitochondrial respiration), and 2-deoxyglucose (2-DG) (a specific inhibitor of glycolysis). The knockdown of miR-542-3p (miR-542-3p) significantly reduced the levels of ECAR compared to that in cells treated with microRNA hairpin inhibitor negative control (Control) (Figure 2A). Notably, the knockdown of miR-542-3p significantly suppressed the levels of ECAR in response to glucose relative to that in control (Figure 2A). In addition, the levels of ECAR in response to oligomycin were reduced by the knockdown of miR-542-3p compared to that in control (Figure 2A). Next, we investigated the molecular target of miR-542-3p in the regulation of glycolytic activity in human glioma cells. We examined whether miR-542-3p could regulate the expression of glycolytic enzymes in the glycolysis pathway in human glioma cells (Figure 2B). Similarly to the levels of ECAR, the knockdown of miR-542-3p significantly reduced the protein levels of HK2 compared to control (Figure 2B). In addition, the knockdown of miR-542-3p decreased the protein levels of LDH-A relative to that of control (Figure 2B). In contrast, the protein levels of other glycolytic enzymes such as phosphofructokinase (PFKP) and pyruvate kinase isozymes M2 (PKM2) were unchanged by miR-542-3p knockdown (Figure 2B). Consistently, over-expression of miR-542-3p (miR-542-3p KO) significantly increased the glycolytic activity in response to glucose compared to that in control (Control) (Figure 2C). Moreover, over-expression of miR-542-3p increased the protein levels of HK2 and LDH-A relative to that in cells treated with microRNA mimic negative control (Control) (Figure 2D). The protein levels of other glycolytic enzymes including PFKP and PKM2 were not changed by the over-expression of miR-542-3p (Figure 2D). These results suggest that miR-542-3p regulates the HK2-mediated high glycolytic phenotype in human glioma cells.

### 3.3. The Levels of miR-542-3p Were Elevated in Patients with High-Grade Gliomas

Next, we investigated whether the levels of miR-542-3p were elevated in patients with gliomas. We analyzed the levels of miR-542-3p in the glioma tissues from patients with low- and high-grade gliomas. To evaluate the levels of miR-542-3p in glioma cells, we measured the levels of miR-542-3p in glioma tissues from patients with low- and high-grade gliomas using in situ hybridization of miR-542-3p (Figure 3A). ISH analysis revealed that the intensity of miR-542-3p-positive staining was significantly elevated in patients with high-grade gliomas (grade III and IV) compared to that in patients with low-grade gliomas (grade I and II) (Figure 3A,B). Next, we investigated the role of miR-542-3p in the prognosis of patients with gliomas. We examined the correlation between the levels of miR-542-3p and the survival rate of patients with gliomas. We performed the Kaplan–Meier estimation test and Cox-regression analysis of patients with low and high levels of miR-542-3p using the datasets from The Cancer Genome Atlas (TCGA) (Figure 3C). The high levels of miR-542-3p were associated with poor survival rate in the GBM and LGG datasets from the TCGA by median survival rate with log-rank test (Log-rank *p* = 0.0037, Cox-regression *p* = 0.004, hazard ratio: 1.7 (Confidence interval: 1.18–2.43), and C-index with Cox-regression analysis: 0.6 (standard error: 0.02)) (Figure 3C). These results suggest that the levels of miR-542-3p were elevated and were associated with poor survival rates in patients with gliomas.

### 3.4. The Levels of HK2 Were Elevated and Positively Correlated with the Levels of miR-542-3p in Patients with Glioma

We next investigated whether miR-542-3p could regulate the levels of HK2 in patients with gliomas. We first analyzed the protein levels of HK2 in patients with low- and high-grade gliomas. Immunofluorescence staining showed the protein levels of HK2 were elevated in glioma tissues of patients with high-grade gliomas (G3 and G4) compared to that of patients with low-grade gliomas (G1 and G2) (Figure 4A). The intensity of HK2-positive staining was elevated in tumor tissues of patients with high-grade gliomas (G3 and G4) relative to that in patients with low-grade gliomas (G1 and G2) (Figure 4B). Moreover, the number of cells with HK2 positive staining was significantly increased in patients with high-grade gliomas (G3 and G4) relative to that in patients with low-grade gliomas (G1 and G2) (Figure 4C). Next, we investigated the correlation between the levels of the *HK2* gene and miR-542-3p levels in patient with gliomas. There was a weak positive-correlation between miR-542-3p and HK2 levels in the analysis of the GBM and LGG datasets from the TCGA (Figure 4D). These results suggest that the levels of HK2 were elevated and were positively correlated with the levels of miR-542-3p in patients with gliomas.

## 4. Discussion

Our results demonstrate that miR-542-3p contributes to the HK2-mediated high glycolytic phenotype in human glioma cells. We suggest that miR-542-3p regulates HK2-mediated high glycolytic activity. Knockdown of miR-542-3p suppresses cellular proliferation. Moreover, the levels of miR-542-3p and HK2 are elevated in patients with gliomas. The levels of miR-542-3p were positively correlated with the levels of HK in the TCGA. The high levels of miR-542-3p contribute to poor prognosis in patients with gliomas. Our findings suggest that miR-542-3p regulates the high glycolytic phenotype via HK2-mediated glycolysis in gliomas.

The high glycolytic phenotype is a critical metabolic phenotype in high-grade glioma such as GBM [27,28,29]. Expression of genes such as HK2 in the glycolysis pathway is essential for the growth of GBM [30]. In particular, the expression of HK2 is higher in the proliferating and apoptosis-resistant regions of GBM than it is in the invading peripheral region [8]. The elevation of HK2 was also found in the most common malignant brain tumors [31]. High levels of glycolysis have been associated with cancer cell proliferation and cellular signaling and functions [32]. Glycolysis produces biosynthetic materials to support cellular proliferation [33]. Consistent with previous study, our results showed that the levels of HK2 were elevated in glioma tissues from patients with high-grade gliomas. The elevation of HK2 increased the glycolytic activity in response to glucose in human glioma cells. Our results suggest that high levels of HK2 are important for the high glycolytic phenotype of glioma cells in high-grade glioma.

Glycolytic activity is regulated by the expression or the enzyme activity of glycolytic enzymes in the glycolytic pathway. During the initiation and progression of tumors, various transcription factors and miRNAs contribute to the regulation of HK2 expression [34,35]. In previous study, hypoxia-inducible factor-1α (HIF-1α) was associated with the regulation of HK2 expression [36,37]. The AKT signaling pathway regulates the transcription of the *HK2* gene. The inhibition of PI3K, an upstream kinase of AKT, and mTORC1, a downstream kinase of AKT, suppresses the expression levels of HK2 [38,39,40,41,42,43,44,45]. In addition, miRNAs regulate the expression of HK2 in cancer cells [46]. miR-155 induces HK2 by transcriptional activation by the signal transducer and activator of transcription 3 (STAT3) as a transcriptional activator or post-transcriptional regulation via repression of miR-143a as a negative regulator of HK2 in human breast cancer cells [46]. Based on these previous studies, STAT3 or miR-143a might be a molecular target in the regulation of HK2 expression in human glioma cells.

Consistent with previous studies, we showed that miR-542-3p promotes the expression of HK2 and HK2-dependent high glycolytic activity in human glioma cells. In addition, the inhibition of miR-542-3p suppressed cellular proliferation in human glioma cells. Our results suggest that miR-542-3p plays an oncogenesis role through the regulation of glycolytic activity in gliomas. Although our results showed a role of miRNAs as an oncogene, previous studies suggested that miRNAs including miR-125a and miR-143 play a tumor suppressor role through inhibition of glucose metabolism in several types of tumors [46,47]. miR-125a and miR-143 negatively regulate the levels of HK2 in hepatocellular carcinoma cells (HCC) and breast cancer cells. Since miRNAs could regulate various target genes in multiple tumor microenvironment conditions, further study is needed on the regulation and role of miR-542-3p in other types of cancer cells such as HCC and breast cancer cells.

Because GBM is the most common malignant high-grade glioma, early diagnosis of GBM is the most effective approach for better patient outcomes [48,49,50,51]. Recent studies showed that different levels of miRNAs were found in brain tissues and peripheral blood from patients with GBM [52]. Since some highly stable miRNAs circulate in the cerebrospinal fluid (CSF) and blood in patients [48,49,50,51,52], miRNAs might be novel diagnostic markers for patients with gliomas. In our study, we showed the elevation of miR-542-3p in tumor tissues of patients with high-grade gliomas. Moreover, we showed the levels of miR-542-3p positively correlated with HK2 levels in analysis of the GBM and LGG datasets from the TCGA. Since high levels of HK2 lead to the high glycolytic phenotype in GBM, the quantification of miR-542-3p levels could be a prognosis marker for the high glycolysis metabolic phenotype of high-grade gliomas, including GBM. Since our study found the elevation of miR-542-3p levels in tumor tissues of patients with gliomas, further study for the quantification of miR-542-3p levels in the blood and CSF in patients with gliomas would need to be studied.

Here, we suggest that miR-542-3p might be a critical factor for the high glycolytic phenotype in glioma. We also suggest the potential role of miR-542-3p as a metabolic alteration regulator in glioma.

## 5. Conclusions

Our findings suggest that miR-542-3p contributes to the HK2-mediated high glycolytic phenotype in human glioma cells. In addition, our results suggest that miR-542-3p is associated with poor prognosis in patients with gliomas. Conclusively, our findings suggest that miR-542-3p might be a critical molecule for the HK2-mediated high glycolytic phenotype in glioma.

## Figures and Tables

**Figure 1 genes-12-00633-f001:**
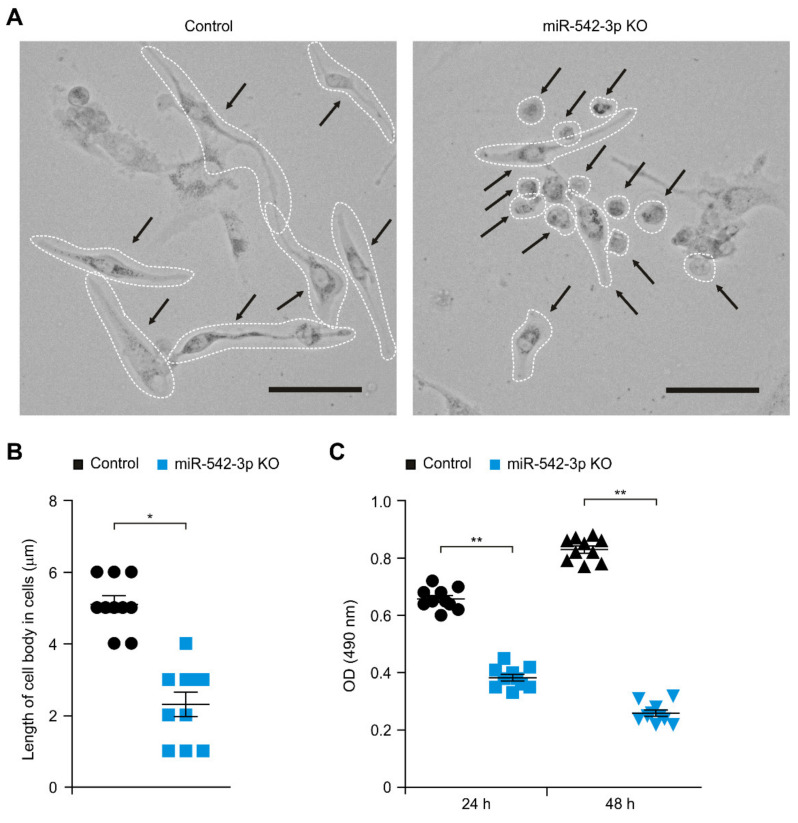
The knockdown of miR-542-3p suppresses cellular proliferation in human glioma cells. (**A**) Representative cellular morphology images and (**B**) quantification of lengths of cell bodies of control (Control) and miR-542-3p knockdown (miR-542-3p KO) human glioma U87MG cells. (*n* = 10 images per individual subject, *n* = 3 per group). Scale bars, 20 μm. Data are representative from three independent experiments. Data are mean ± SD. * *p* < 0.05 by the two-tailed Student’s *t*-test. (**C**) Cell proliferation assay in control (Control) and miR-542-3p knockdown (miR-542-3p KO) human glioma U87MG cells. (*n* = 10 images per individual subject, *n* = 3 per group). Data are mean ± SD. ** *p* < 0.01 by the two-tailed Student’s *t*-test.

**Figure 2 genes-12-00633-f002:**
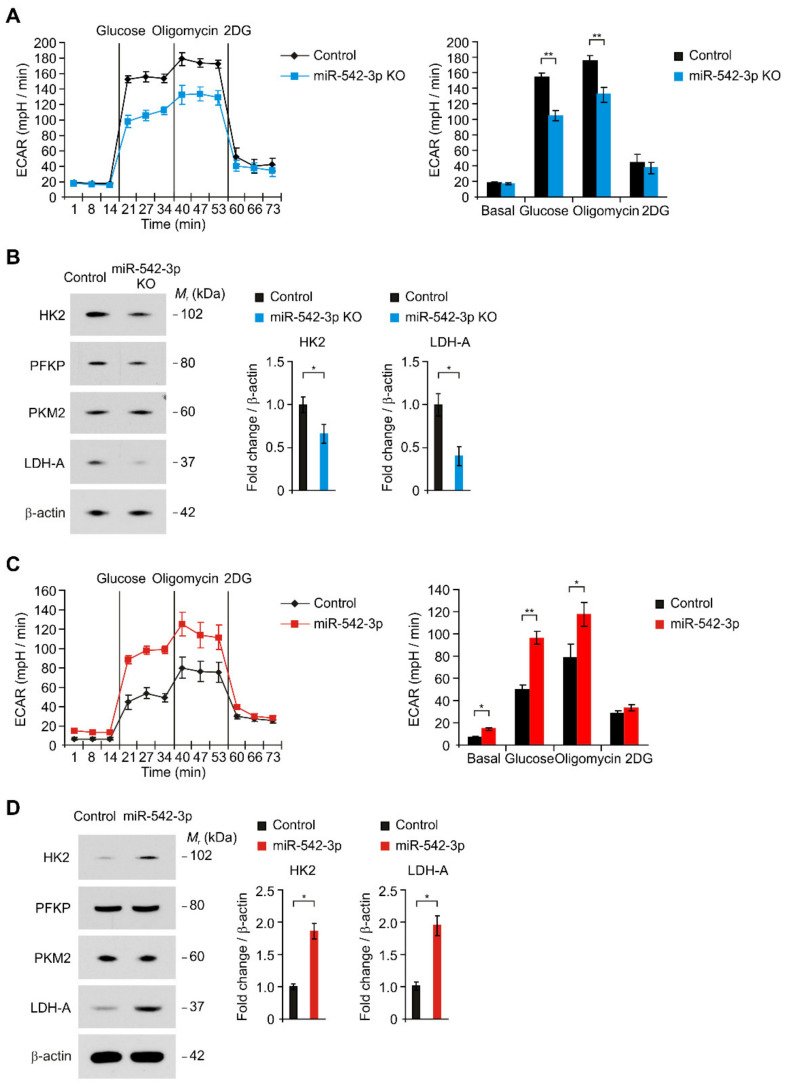
miR-542-3p regulates the HK2-mediated high glycolytic phenotype in human glioma cells. (**A**) The ECAR levels in response to glucose, oligomycin, and 2DG (left) and quantification of ECAR levels (right) in microRNA hairpin inhibitor negative control treated (Control) and miR-542-3p knockdown (miR-542-3p KO) human glioma U87MG cells. Representative data are determined from three independent experiments. Data are mean ± SEM. ** *p* < 0.01; * *p* < 0.05 by two-tailed Student’s *t*-test. (**B**) Representative immunoblot analysis for HK2, PFKP, PKM2, and LDH-A protein levels (left) and quantification for HK2 and LDH-A protein levels (right) in microRNA hairpin inhibitor negative control treated (Control) and miR-542-3p knockdown (miR-542-3p KO) human glioma U87MG cells. In immunoblots, the levels of β-actin were used as loading control. Representative data are determined from three independent experiments. Data are mean ± SD. * *p* < 0.05 by the two-tailed Student’s *t*-test. (**C**) The ECAR levels in response to glucose, oligomycin, and 2DG (left) and quantification of ECAR levels (right) in microRNA mimic negative control treated (Control) and miR-542-3p over-expressed (miR-542-3p) human glioma U87MG cells. Representative data are determined from three independent experiments. Data are mean ± SEM. ** *p* < 0.01 by two-tailed Student’s *t*-test. (**D**) Representative immunoblot analysis for HK2, PFKP, PKM2, and LDH-A protein levels (left) and quantification for HK2 and LDH-A protein levels (right) in microRNA mimic negative control treated (Control) and miR-542-3p over-expressed (miR-542-3p) human glioma U87MG cells. In immunoblots, the levels of β-actin were used as loading control. Representative data are determined from three independent experiments. Data are mean ± SD. * *p* < 0.05 by the two-tailed Student’s *t*-test.

**Figure 3 genes-12-00633-f003:**
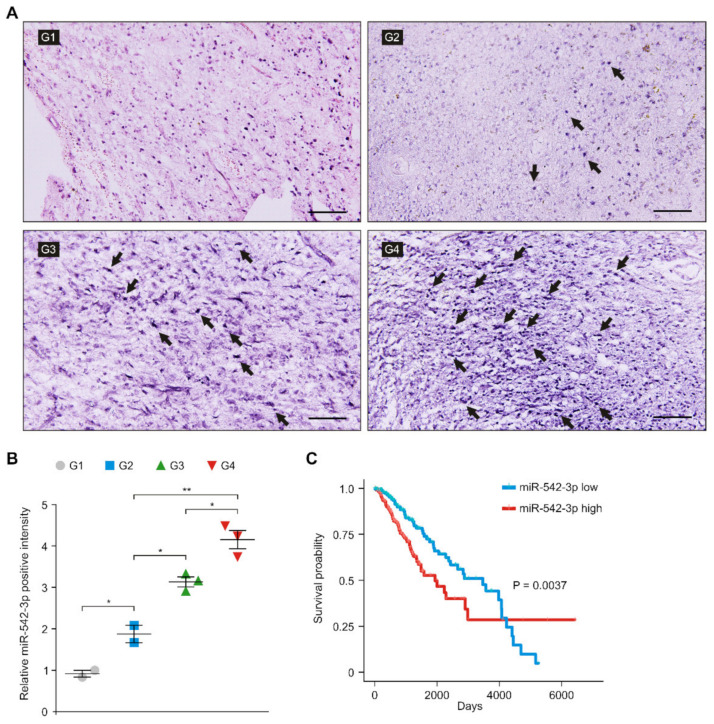
The levels of miR-542-3p were elevated in patients with high-grade gliomas. (**A**) Representative ISH images of miR-542-3p expression in tissues from patients with low- and high-grade gliomas (G1; grade I, G2; grade II, G3; grade III, G4; grade IV) showing miR-542-3p positive (purple) in glioma cells (*n* = 10 images per individual subject, *n* = 3 per group). Scale bars, 100 μm. Black arrows indicate miR-542-3p positive cells. (**B**) Quantification of miR-542-3p positive glioma cells from ISH images in tissues from low- and high-grade patients with gliomas (G1, G2, G3, G4) (*n* = 10 images per individual subject, *n* = 3 per group). Data are mean ± SD. **, *p* < 0.01; *, *p* < 0.05 by Student’s two-tailed *t*-test. (**C**) Survival curve of patients by median survival rate with log-rank test with low levels (blue) and high levels (red) of miR-542-3p (*n* = 690) was determined from the GBM and LGG datasets from the TCGA. *p* = 0.0037 by two-sided and one-sided Kaplan–Meier estimation tests.

**Figure 4 genes-12-00633-f004:**
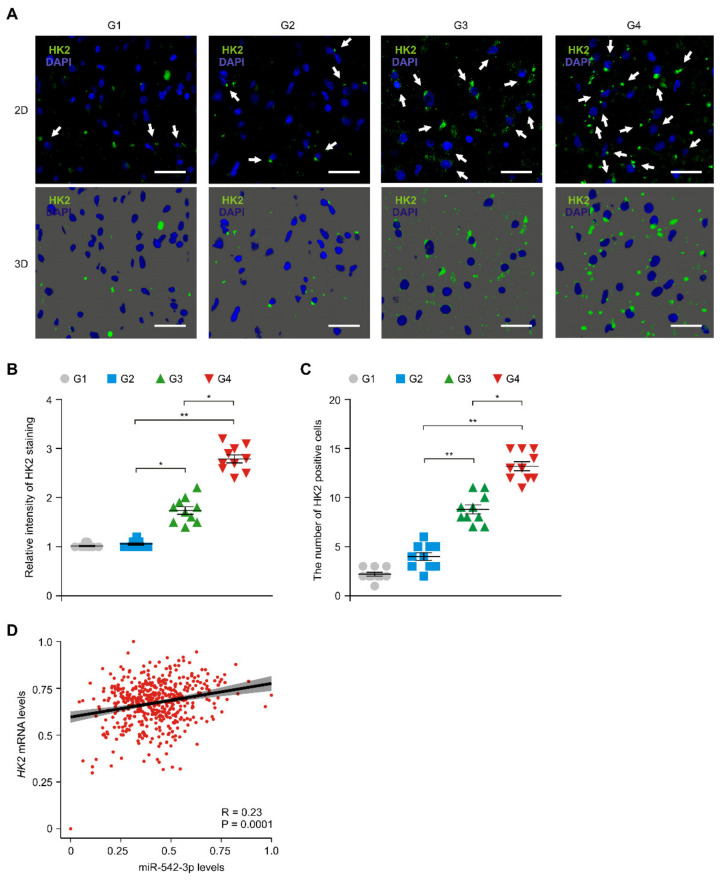
The levels of HK2 were elevated and were positively correlated with the levels of miR-542-3p in patients with gliomas. (**A**) Representative immunofluorescence 2D (top) and 3D (bottom) images of HK2 protein expression in glioma cells of tissues from patients with low- and high-grade gliomsa (G1, G2, G3, G4) showing HK2 (green) in glioma cells (*n* = 10 images per individual subject, *n* = 3 per group). Blue indicates DAPI-stained nuclei in cells. Scale bars, 20 μm. White arrows indicate HK2 positive cells. (**B**) Quantification of relative HK2 positive staining from immunofluorescence images in tissues from patients with low- and high-grade gliomas (G1, G2, G3, G4) (*n* = 10 images per individual subject, *n* = 3 per group). Data are mean ± SD. **, *p* < 0.01; *, *p* < 0.05 by Student’s two-tailed *t*-test. (**C**) Quantification of HK2 positive glioma cells from immunofluorescence images in tissues from patients with low- and high-grade gliomas (G1, G2, G3, G4) (*n* = 3 per group, *n* = 10 per area of individual subject). Data are mean ± SD. **, *p* < 0.01; *, *p* < 0.05 by Student’s two-tailed *t*-test. (**D**) A Spearman correlation coefficient analysis and linear regression analysis to determine the relationship between HK2 and miR-542-3p in patients with gliomas in the GBM and LGG datasets from the TCGA. Data are mean ± standard deviation (SD). R = 0.23, *p* = 0.0001 by Spearman correlation coefficient test.

## Data Availability

Not acceptable.

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
