# Peer review of "miR-542-3p Contributes to the HK2-Mediated High Glycolytic Phenotype in Human Glioma Cells"

_genes, 2021, doi:10.3390/genes12050633_

Round 1

Reviewer 1 Report

In this manuscript, the roll of miR-542-3p and HK2-mediated glycolytic activity in human glioma cells has been evaluated. Authors suggest that miR-542-3p is associated with poor prognosis in glioma.

The methodology is clear and results are presented in a high quality manner. Their conclusions can contribute to expand the research on this field.

Just a couple of minor concerns,

  1. Figure 3 C, shows KM curves, but in the high group has better survival probabilities than low group, contrary to that mentioned in the text, is that an error?
  2. I suggest to perform a Cox regression to provide HR and C-index and not just KM curves.
  3. Figure 4 D, shows a scatter plot between miR-542-3p and HK2, although significant, the correlation is weak. Should be mention in the text. Has the author tried to perform a linear regression between these variables ???

Author Response

Response to Genes Reviewer 1 Comments

In this manuscript, the roll of miR-542-3p and HK2-mediated glycolytic activity in human glioma cells has been evaluated. Authors suggest that miR-542-3p is associated with poor prognosis in glioma.

The methodology is clear and results are presented in a high quality manner. Their conclusions can contribute to expand the research on this field.

Just a couple of minor concerns,

Reviewer’s Comment 1

  1. Figure 3 C, shows KM curves, but in the high group has better survival probabilities than low group, contrary to that mentioned in the text, is that an error?

I suggest to perform a Cox regression to provide HR and C-index and not just KM curves.

Response 1

We evaluated the statistical significance of median survival rate with log-rank test between high and low expression group of miR-542-3p. The statistical significance is indicated by the median survival rate at the time point of 2000 days. The association between the high levels of miR-542-3p and poor survival rate is correct. As reviewer’s comment, we performed an additional statistical analysis for Cox regression. We provided hazard ratio (HR) and C-index by Cox regression analysis. Also, we described the method of Cox regression in Materials and Methods section.

The following text has been added to Page 9. Line 305 and Page 5. Line 213:

Page 9. Line 305 “We performed the Kaplan-Meier estimation test and Cox-regression analysis of patients with low and high levels of miR-542-3p using the dataset from The Cancer Genome Atlas (TCGA) (Figure 3C). The high levels of miR-542-3p were associated with poor survival rate in GBM and LGG dataset from TCGA by median survival rate with log-rank test (Log-rank p = 0.0037, Cox-regression p = 0.004, hazard ratio: 1.7 [Confidence interval: 1.18 - 2.43], and C-index with Cox-regression analysis: 0.6 [standard error: 0.02]) (Figure 3C).”

Page 5. Line 213 “Linear regression was performed to confirm the relationship between miR-542-3p and HK2 expression level. Kaplan-Meier estimation and Cox-regression analysis were per-formed to analyze prognosis of genes with overall survival data. Concordance index was calculated to provide accuracy with survival proportion.”

Reviewer’s Comment 2

  1. Figure 4 D, shows a scatter plot between miR-542-3p and HK2, although significant, the correlation is weak. Should be mention in the text. Has the author tried to perform a linear regression between these variables ???

Response 2

We agree with reviewer’s comment for the weak correlation between miR-542-3p and HK2. As reviewer’s comment, we described that there was a weak positive-correlation between miR-542-3p and HK2. As reviewer’s comment, we performed linear regression analysis between miR-542-3p and HK2. We obtained statistically significant result (p < 0.001).

The following text has been added to Page 11. Line 339:

Page 11. Line 339 “There was a weak positive-correlation between miR-542-3p and HK2 levels in analysis of GBM and LGG dataset from TCGA (Figure 4D).”

Reviewer 2 Report

In the manuscript the authors evaluated the meaning of miR-542-3p in glioma cells. Based on clinical data and TGCA database they showed overexpression of this particular miRNA in high-grade gliomas, including GBM. Moreover, high expression of miR-542-3p was linked to HK2-mediated glycolytic phenotype of glioma cells.

Broad comments

In general, the study was well designed and in vitro and in vivo data proved the main concept. The results suggest that miR-542-3p overexpression in glioma cells is related to excessive glycolytic activity and is associated with poor prognosis.

However, in connection between miR-542-3p and HK2 overexpression there is a missing mechanistic correlation. In discussion the authors mentioned miR-155 mechanism of HK2 upregulation based on transcriptional activation of STAT3 and repression of miR-143 - a negative regulator of HK2. Thus, miR-542-3p, if enhance HK2 expression, probably possess some molecular target(s) controlling HK2 expression. This should be at least theoretically discussed in the manuscript.

Specific comments

In figure 2. glycolytic phenotype was analyzed. miR-542-3p KO reduced, and its overexpression enhanced extracellular acidification - comparing to the control. Unfortunately, the ECAR values for control cells in figure 2A and 2C differs a lot, e.g. in glucose environment in 2A ECAR is app. 150, and in 2C app. 50 mpH/min. In addition, HK2 band for control in 2D is much weaker comparing to 2B control, and in 2B the band is similar to β-actin, where in 2D they are totally different. Why so much differences between results are observed? Is there any differences between both controls that can affect results?

Text editing is also required. Some examples:

line 18-19, 51 such as including --> such as OR including

line 47 to maintenance cellular --> to maintain of cellular

line 68 miRNA play as tumor - please use other verb

line 110 10 % (vol/vol) refers to antibiotics? 

line 145 12.5 nm? probably nM

line 148 were performed blocking with blocking solution – please rewrite

line 217 We next – please rewrite (it not suits at the beginning of results)

line 260 miR-542-3p KO? - rather overexpression

line 349-352 – please rewrite to avoid stylistic repeats, e.g. each time ...in human glioma cells

line 363 during glycolysis, glycolysis produces – please rewrite 

Author Response

Response to Genes Reviewer 2 Comments

In the manuscript the authors evaluated the meaning of miR-542-3p in glioma cells. Based on clinical data and TGCA database they showed overexpression of this particular miRNA in high-grade gliomas, including GBM. Moreover, high expression of miR-542-3p was linked to HK2-mediated glycolytic phenotype of glioma cells.

Reviewer’s Comment 1

Broad comments

In general, the study was well designed and in vitro and in vivo data proved the main concept. The results suggest that miR-542-3p overexpression in glioma cells is related to excessive glycolytic activity and is associated with poor prognosis.

However, in connection between miR-542-3p and HK2 overexpression there is a missing mechanistic correlation. In discussion the authors mentioned miR-155 mechanism of HK2 upregulation based on transcriptional activation of STAT3 and repression of miR-143 - a negative regulator of HK2. Thus, miR-542-3p, if enhance HK2 expression, probably possess some molecular target(s) controlling HK2 expression. This should be at least theoretically discussed in the manuscript.

Response 1

As reviewer’s comment, we provided additional descriptions for the possible molecular targets such as STAT3 or miR-143a that could regulate HK2 expression in human glioma cells in discussion section.

The following text has been added to Page 12. Line 391:

Page 12. Line 391 “Based on these previous studies, STAT3 or miR-143a might be a molecular target in the regulation of HK2 expression in human glioma cells.”

Reviewer’s Comment 2-1

Specific comments

In figure 2. glycolytic phenotype was analyzed. miR-542-3p KO reduced, and its overexpression enhanced extracellular acidification - comparing to the control. Unfortunately, the ECAR values for control cells in figure 2A and 2C differs a lot, e.g. in glucose environment in 2A ECAR is app. 150, and in 2C app. 50 mpH/min.

Response 2-1

As reviewer’s comment, there was a difference in the ECAR values for control cells between Figure 2A and 2C. Since we used the different number of cells in two independent experiments belong to Figure 2A and 2C, there was the difference of ECAR values in control cells. To show the precise relative results in two groups, we used the different number of cells in two independent experiments. In miR-542-3p knockdown experiments, we used 5 x 104 cells/well. In miR-542-3p knockdown experiments, we used 2.5 x 104 cells/well. We provided additional information for cell number in Materials and Methods.

The following text has been added to Page 4. Line 176:

Page 4. Line 176 “For the glycolytic function assay, human glioma cells (5 x 104 cells/well) were plat-ed on XF96e cell culture microplates for miR-542-3p knockdown experiments (101085-004, Agilent Technologies, Inc., Santa Clara, CA, USA). Also, human glioma cells (2.5 x 104 cells/well) were plated on XF96e cell culture microplates for miR-542-3p over-expression experiments.”

Reviewer’s Comment 2-2

In addition, HK2 band for control in 2D is much weaker comparing to 2B control, and in 2B the band is similar to β-actin, where in 2D they are totally different. Why so much differences between results are observed? Is there any differences between both controls that can affect results?

Response 2-2

To show the precise relative comparison in two groups, we need to use the different amounts of protein from cell lysates in immune blot analysis. Since we need to show the reduction of HK2 levels in Figure 2B, we relatively applied large amounts of protein in this comparison. On the other hands, we relatively applied small amounts of protein when we need to show the increase of HK2 levels in Figure 2D. That is the reason for the different HK2 levels in control group between Figure 2B and Figure 2D.

Reviewer’s Comment 3

Text editing is also required. Some examples:

line 18-19, 51 such as including --> such as OR including

Response 3-1

As reviewer’s comment, we removed ‘including’.

line 47 to maintenance cellular --> to maintain of cellular

Response 3-2

As reviewer’s comment, we revised as ‘is important for the maintenance of cellular’.

line 68 miRNA play as tumor - please use other verb

Response 3-3

As reviewer’s comment, we revised as ‘various miRNAs have a role’.

line 110 10 % (vol/vol) refers to antibiotics?

Response 3-4

As reviewer’s comment, we removed ‘10 % (vol/vol)’. It was error. We provided the concentration of antibiotics.

line 145 12.5 nm? probably nM

Response 3-5

As reviewer’s comment, we revised. nM is correct.

line 148 were performed blocking with blocking solution – please rewrite

Response 3-2

As reviewer’s comment, we revised as “slides were incubated with blocking solution”.

line 217 We next – please rewrite (it not suits at the beginning of results)

Response 3-6

As reviewer’s comment, we removed ‘next’.

line 260 miR-542-3p KO? - rather overexpression

Response 3-7

As reviewer’s comment, we revised as ‘the knockdown of miR-542-3p’.

line 349-352 – please rewrite to avoid stylistic repeats, e.g. each time ...in human glioma cells

Response 3-8

As reviewer’s comment, we removed ‘in human glioma cells’ in the repeat parts.

line 363 during glycolysis, glycolysis produces – please rewrite

Response 3-9

As reviewer’s comment, we removed ‘during glycolysis’.

As reviewer’s comment, we completed spell and grammar check.

Also, we will have the English editing by the journal.

Round 2

Reviewer 2 Report

After revision and modifications according to the previous suggestions, the manuscript is properly improved. I do not have additional comments.

Author Response

Response to Genes Reviewer 2 Comments

Reviewer’s Comment 1

After revision and modifications according to the previous suggestions, the manuscript is properly improved. I do not have additional comments.

Response 1

Thank you for your suggestions. Your suggestions helped to improve our paper.
